# Epidemiology and Molecular Transmission Characteristics of HIV in the Capital City of Anhui Province in China

**DOI:** 10.3390/pathogens10121554

**Published:** 2021-11-29

**Authors:** Shan Zheng, Jianjun Wu, Zhongwang Hu, Mengze Gan, Lei Liu, Chang Song, Yanhua Lei, Hai Wang, Lingjie Liao, Yi Feng, Yiming Shao, Yuhua Ruan, Hui Xing

**Affiliations:** 1State Key Laboratory of Infectious Disease Prevention and Control, National Center for AIDS/STD Control and Prevention, Chinese Center for Disease Control and Prevention, Beijing 102206, China; zhengshan0926@163.com (S.Z.); ganmengze@126.com (M.G.); cdcliulei@163.com (L.L.); songchang604@163.com (C.S.); liaolj@chinaaids.cn (L.L.); fengyi@chinaaids.cn (Y.F.); yshao@bjmu.edu.cn (Y.S.); ruanyuhua92@163.com (Y.R.); 2Anhui Provincial Center for Disease Control and Prevention, Hefei 230601, China; wujianjun823@126.com; 3Hefei Center for Disease Control and Prevention, Hefei 230061, China; hzw1100@126.com (Z.H.); 13839084827@163.com (Y.L.); swan_qi@163.com (H.W.)

**Keywords:** HIV, molecular network, transmission cluster, effective reproductive number

## Abstract

Hefei, Anhui province, is one of the cities in the Yangtze River Delta, where many people migrate to Jiangsu, Zhejiang and Shanghai. High migration also contributes to the HIV epidemic. This study explored the HIV prevalence in Hefei to provide a reference for other provinces and assist in the prevention and control of HIV in China. A total of 816 newly reported people with HIV in Hefei from 2017 to 2020 were recruited as subjects. HIV subtypes were identified by a phylogenetic tree. The most prevalent subtypes were CRF07_BC (41.4%), CRF01_AE (38.1%) and CRF55_01B (6.3%). Molecular networks were inferred using HIV-TRACE. The largest and most active transmission cluster was CRF55_01B in Hefei’s network. A Chinese national database (50,798 sequences) was also subjected to molecular network analysis to study the relationship between patients in Hefei and other provinces. CRF55_01B and CRF07_BC-N had higher clustered and interprovincial transmission rates in the national molecular network. People with HIV in Hefei mainly transmitted the disease within the province. Finally, we displayed the epidemic trend of HIV in Hefei in recent years with the dynamic change of effective reproductive number (Re). The weighted overall Re increased rapidly from 2012 to 2015, with a peak value of 3.20 (95% BCI, 2.18–3.85). After 2015, Re began to decline and remained stable at around 1.80. In addition, the Re of CRF55_01B was calculated to be between 2.0 and 4.0 in 2018 and 2019. More attention needs to be paid to the rapid spread of CRF55_01B and CRF07_BC-N strains among people with HIV and the high Re in Hefei. These data provide necessary support to guide the targeted prevention and control of HIV.

## 1. Introduction

As of 2020, there were 37.7 million people living with HIV worldwide [1]. The prevalence and transmission of HIV remains a huge challenge to public health in China. There has been many reports on HIV in key endemic areas, but HIV research in characteristically large cities in China is also very important and needed. HIV infections in Anhui province originated with earlier commercial blood donations [2]. Gradually, sexual transmission has become the main transmission route of people with HIV in Anhui province [3]. Hefei, the capital of Anhui province, is a dual-node city under the Belt and Road Initiative and the Yangtze River Economic Belt. It is an important central city and comprehensive transportation hub in eastern China. The current HIV transmission route mainly involves homosexual transmission in Hefei. It has high population mobility and floating populations may promote the transmission and prevalence of HIV. Studying the prevalence and transmission of HIV in Hefei can ensure effective intervention in active transmission clusters and reduction in the transmission of HIV.

Recent studies on the application of new techniques and methods in molecular epidemiology have shown the power of various HIV gene sequence and analysis tools in improving transmission detection and intervention guidance [4,5,6]. The HIV molecular network is a type of transmission analysis based on the group model, which considers a potential transmission relationship between the infected people connected together. We can identify active transmission clusters that require critical attention and intervention through molecular networks [7,8,9]. The effective reproductive number (Re) is often used to describe the dynamics of transmission during an epidemic. Phylodynamic modeling can combine genetic modeling with epidemiological modeling to quantify Re in HIV infections [10,11,12].

In this study, we explored the prevalence and transmission of newly reported HIV infections in Hefei using the molecular network and estimated the Re of different subtypes. The results provide a basis for accurate prevention and control of HIV transmission and a reference for research in other provinces and cities.

## 2. Results

### 2.1. Demographic Characteristics and Subtype Profile of the Study Population

A total of 816 newly reported HIV patients in Hefei, Anhui Province from 2017 to 2020 were included in this study. According to the local center for Disease Control and Prevention, there were 1468 newly reported HIV infections in Hefei from 2017 to 2020. The sampling ratio was 55.6% (816/1468). Among the 816 patients, 69.3% (565/816) were under the age of 50; 76.8% (627/816) were male; 61.9% (505/816) and 21.8% (178/816) of the patients were infected through homosexual and heterosexual contact, respectively; 46.4% (379/816) were single; 56.6% (462/816) had obtained a high school and above education, and 67.4% (550/816) had a CD4^+^ T cell count of <500 cells/μL (Table 1).

CRF07_BC subtype presented two distinct clusters in the phylogenetic tree: CRF07_BC-N, the subcluster that mainly spread among men who have sex with men (MSM); CRF07_BC-O, the subcluster that mainly spread in heterosexuals and injecting drug users (IDUs) [13]. A total of seven clusters were identified for the CRF01_AE subtype (clusters 1–7), plus an additional cluster classified as CRF01_AE-others [14]. Of the 816 total cases, 41.4% (338/816) were CRF07_BC, including 34.9% (285/816) CRF07_BC-N and 6.5% (53/816) CRF07_BC-O. A total of 38.1% (311/816) were CRF01_AE, comprised of six subclusters: 2.8% (23/816) CRF01_AE-cluster 1, 0.9% (7/816) CRF01_AE-cluster 2, 0.7% (6/816) CRF01_AE-cluster 3, 24.4% (199/816) CRF01_AE-cluster 4, 8.1% (66/816) CRF01_AE-cluster 5, and 1.2% (10/816) CRF01_AE-others. A total of 6.3% (51/816) were CRF55_01B; CRF08_BC and subtype B accounted for 3.4% (28/816) each. In addition, CRF67_01B (n = 10), CRF68_01B (n = 16), CRF79_0107 (n = 4), CRF65_cpx (n = 2), and URFs (n = 28) were also identified (Figure 1). CRF67_01B and CRF68_01B had also become popular clusters in Hefei. The HIV *pol* fragments of 28 cases of URF were found to be the recombinant strains of CRF01_AE and B or C subtype by JPHMM and RIP. The prevalent HIV strains in Hefei were subtypes and subclusters that were transmitted mainly among MSM.

### 2.2. Molecular Network Analysis and Active Transmission Clusters

Under the threshold of 0.5% genetic distance, 27.1% (221/816) of the sequences from 69 clusters were enrolled in the molecular network. The largest cluster consisted of 10.0% (22/221) sequences. A molecular network diagram of Hefei is shown in Figure 2. 

A total of 816 plasma samples were eligible for HIV limiting-antigen avidity enzyme immunoassay (LAg-Avidity EIA), and 22.1% (180/816) of them were recently infected. Based on newly reported HIV infections in 2017–2019, a transmission cluster with at least three recent HIV infections in 2020 is defined as an active transmission cluster. There were three active transmission clusters: the largest cluster was CRF55_01B, followed by CRF01_AE-cluster 4, and CRF01_AE-cluster 5. There were seven clusters containing two recent infections in 2020, including two clusters of CRF07_BC-N and five clusters of CRF01_AE-cluster 4.

### 2.3. Analysis of Interprovincial Transmission Characteristics

After removing the repeated sequences, 50,798 sequences were obtained from the LANL and China CDC databases as of 30 June 2020. Dataset B covered 31 provinces in China from 2000 to 2020 (Appendix A). Under the threshold of 0.5% genetic distance, 35.8% (292/816) of the sequences from Hefei were enrolled in the molecular network, which increased by 8.7% compared with the Hefei’s network. The interprovincial transmission rate in Hefei was 9.8% (92/943). We evaluated factors associated with clustering and interprovincial transmission. After adjusting for other factors, the CRF55_01B and CRF07_BC-N strains entered the network more easily, and their interprovincial transmission rates were higher (Table 2 and Table 3).

Sequences from Hefei existed in 140 clusters and formed 1268 links with the sequences from 23 provinces, of which 67.5% (856/1268) were linked to Hefei itself, 10.8% (137/1268) were related to other cities in Anhui province, and 21.7% (275/1268) were linked to other provinces. The links between Hefei and each province are shown in Figure 3.

### 2.4. Estimating the Re of the Main Prevalent Subtypes and Subclusters

The Re values of several major epidemic subtypes and subclusters were estimated. The Re of CRF01_AE-cluster 4 reached its peak during 2013–2015 (Re = 3.85; 95% BCI, 2.12–5.88); CRF01_AE-cluster 5 peaked in 2016 (Re = 3.78; 95% BCI, 0.92–5.82). CRF07_BC-N peaked during 2015–2016 (Re = 4.03; 95% BCI, 2.89–5.62). The peak value of Re for CRF55_01B was in 2019 (Re, 4.01, 95% BCI, 1.12–6.21), and remained 2.0–4.0 during 2018 and 2019. Further, we calculated the weighted Re for each year. Overall, the Re in Hefei increased rapidly from 2012 to 2015 and its peak was 3.20 (95% BCI, 2.18–3.85). After 2015, the Re began to decline and remained stable at around 1.80 (Figure 4).

## 3. Discussion

In this study, 816 newly reported people with HIV from 2017 to 2020 were selected to study the prevalence of HIV in Hefei, Anhui Province. We found that MSM were the predominant people with HIV in the Hefei study group. MSM in China are characterized by high mobility, which greatly promotes HIV transmission in different areas [15]. Moreover, MSM has always played a bridge role in the transmission of HIV among different populations [16,17,18]. MSM may have homosexual and heterosexual behaviors at the same time [7]. Therefore, especially in large cities, it is necessary to strengthen the intervention among MSM. The government and relevant departments can raise awareness within MSM groups to prevent high-risk behaviors and encourage them to perform pre-exposure prophylaxis (PrEP) and post-exposure prophylaxis (PEP) through publicity and education. Various subtypes were identified in Hefei, and the main prevalent subtypes were CRF07_BC, CRF01_AE and CRF55_01B. In addition to the common epidemic subtypes, CRF67_01B and CRF68_01B were first reported as epidemics in Hefei. There were also many URFs in Hefei, indicating that many individuals were repeatedly infected with different HIV strains. Frequent recombination of HIV genomes can accelerate the evolution of HIV strains and promote the emergence of HIV strains with high viral fitness [19]. This also suggests that the prevalence of multiple recombinant strains in China is no longer a local epidemic problem. Therefore, it is necessary to strengthen the monitoring of HIV subtypes in some cities, such as Hefei, to avoid the emergence of more URFs, which will likely cause difficulties in prevention and control.

The largest and most active transmission cluster in Hefei was CRF55_01B, as shown in the molecular network. Molecular network analysis of national data showed that CRF55_01B had a higher clustered and interprovincial transmission rate. CRF55_01B, which was first reported in 2013, is a late-emerging recombinant strain in China [20]. However, it has grown rapidly in recent years and has become the fifth most predominant HIV-1 strain in China [21]. As a strain originating with MSM, CRF55_01B is more likely to spread between large cities and across provinces, similar to other strains associated with MSM [22,23]. Our previous research showed that CRF55_01B spreads rapidly from MSM to heterosexual people, and this strain has been found in all provinces, forming transmission clusters in more than half of the provinces. Its rapid transmission may be related to the development of transportation and technology [24]. CRF07_BC-N also showed higher network access and interprovincial transmission rates. CRF07_BC strains originated among IDUs in southwest China and later spread to other provinces of China through IDUs and heterosexuals [25]. In recent years, more and more CRF07_BC strains have been detected in MSM in China [26,27,28]. A study showed that CRF_07BC-N has a greater risk of transmission than CRF07_BC-O [13]. In our unpublished studies, we elucidated the transmission trend and scale of the CRF07_BC epidemic recombinant strain from IDUs to heterosexuals and then to MSM in China. The CRF07_BC-N cluster is the main interprovincial HIV epidemic recombinant strain in China, especially in developed cities. Thus, CRF55_01B and CRF07_BC-N aggregate more readily and spread more widely than the other subtypes. The prevention and control of CRF55_01B and CRF07_BC-N in provincial capitals, such as Hefei, needs to be considered.

In the national molecular network, most of the people with HIV in Hefei were linked internally, and the rest were mainly linked with first-tier cities, such as Beijing, Shanghai, and Guangdong. Owing to the increasingly convenient transportation and population mobility, many research results show that floating populations have become an important factor in the transmission and prevalence of HIV, forming a bridge group for rapid transmission between regions [29]. Floating populations are usually people separated from their spouses, and mainly young adults who are at the peak of sexual activity and generally have high-risk sexual behaviors [30]. These factors make floating populations an important group in HIV transmission. As Hefei is one of the cities in the Yangtze River Delta region, many people from this city migrate to the Jiangsu, Zhejiang, and Shanghai areas. Understanding the prevalence and molecular network transmission of HIV in these big cities can improve intervention accuracy among active transmission groups and reduce the transmission of HIV. The results of this study show that the overall rate of interprovincial transmission in Hefei is not very high. However, interprovincial transmission cannot be ignored due to the rapid movement of the population. In addition, many people who are not in the network may have contact with other provinces.

The dynamic Re for 2012–2020 was estimated. CRF07_BC-N, CRF01_AE-cluster 4, and CRF01_AE-cluster 5 were the subtypes with earlier epidemic times, and the peak of Re appeared around 2013–2016. The weighted Re in Hefei increased rapidly from 2012 to 2015, declined slowly in 2015–2016 and rapidly thereafter, which may be related to the expansion of antiviral therapy in 2016. Some studies have also evaluated the effectiveness of control measures by calculating the dynamic changes in Re [31,32]. CRF55_01B has been an epidemic strain in recent years, and its epidemic time in Hefei is relatively late. However, the Re of CRF55_01B is very high, indicating that it has grown rapidly in Hefei in recent years. The Re results are consistent with the previous conclusion that CRF55_01B is the largest and most active cluster in Hefei. The Re results combined with the LAg-Avidity EIA results showed that the new infection rate in Hefei is still high.

In conclusion, this study involved people with HIV in Hefei as an example and showed that the prevalence of complex and diverse recombinant HIV strains, the rapid spread of the CRF55_01B and CRF07_BC-N strains, and the high Re condition points to the need for greater attention in similar large population cities. This is the first time that an analysis of the association of infected patients in one city with other provinces and cities has been conducted. On the other hand, the very interesting question of whether the active transmission clusters in 2020 are related to COVID-19 city-lockdowns can be further studied. This study has some limitations. All the people with HIV were analyzed according to the reporting place, and there may be some deviation from their actual residence. However, these data will provide data support about the prevalence of HIV in large provincial capital cities. The data and analyses also suggest that the situation may be similar in other provinces and cities where HIV is transmitted primarily among MSM.

## 4. Materials and Methods

### 4.1. Study Population and Design

Newly reported people with HIV from 2017 to 2020 in Hefei were collected by sampling. The inclusion criteria of the study subjects were as follows: (1) age ≥ 18 years; (2) patients with HIV infection who had not received any antiviral treatment from January 2017 to December 2020; and (3) patients who had completed questionnaires and signed informed consent forms (ethics approval number is X140617334).

### 4.2. Laboratory Tests

Plasma samples from the study subjects were collected by the laboratory personnel of the local Center for Disease Control and Prevention (CDC) and transported to the National Center for AIDS/STD Control and Prevention, Chinese Center for Disease Control and Prevention (China CDC) for testing. Viral RNA was extracted using the QIAamp Viral RNA Mini Kit (Qiagen, Hilden, Germany). A nested polymerase chain reaction (PCR) was performed to amplify the HIV *pol* gene fragments at positions 2253–3553 of the international standard strain HXB2 [33]. Rent infections were detected using HIV LAg-Avidity EIA [34]. It includes the preliminary screening test and confirmation test. Samples with ODn less than or equal to 2.0 in the preliminary screening test shall be validated. If the ODn value of the test sample is confirmed to be between 0.4 and 1.5, the sample is recently infected.

### 4.3. HIV Sequence Acquisition and Subtyping

Sequences obtained by Sanger sequencing were spliced using Sequencher 4.10.1 (Gene Codes Corporation, Ann Arbor, MI, USA) and aligned using Mafft 7.037. Sequence quality control was performed by WHO HIVDR QC TOOL (https://sequenceqc.bccfe.ca/who_qc, accessed on 6 July 2021). FastTree 2.1 was used to construct a phylogenetic tree for subtype identification. [35]. The nucleotide substitution model was GTR + G + I, and support values of the nodes were calculated with a Shimodaira Hasegawa-like test. Clusters with a bootstrap value higher than 0.90 (90%) were defined as the same subtype and subclusters. The reference sequences included the major international epidemic strains A–D, F–H, and JK, and the major epidemic recombinant strains from HIV Databases (https://www.hiv.lanl.gov/content/index, accessed on 7 July 2021) and our laboratory. Maximum likelihood trees were imported into FigTree v1.4.3. Unique recombinant forms (URFs) were used to determine recombination breakpoints by JPHMM at GOBICS (http://jphmm.gobics.de/, accessed on 7 July 2021) and RIP (https://www.hiv.lanl.gov/content/sequence/RIP/RIP.html, accessed on 7 July 2021) in the HIV sequence database.

### 4.4. Phylogenetic Analysis and HIV Molecular Network Construction

A molecular network was constructed using HIV TRACE [36]. Aligned *pol* sequences were used to calculate pairwise genetic distances using the Tamura-Nei 93 model [37]. All sequences were longer than 1000 bp, and ambiguous nucleotides were less than 5%. Each patient in the molecular network was represented by a node, and nodes were linked to each other if their pairwise genetic distance was within 0.5% substitutions per site. A threshold of 0.5% was selected to identify transmission relationships over two to three years. HIV *pol* gene region sequences from across China were collected from the HIV sequence database of the Los Alamos National Laboratories (LANL) and China CDC databases to establish dataset B. Dataset B was used to analyze the interprovincial transmission from Hefei. In this study, an interprovincial cluster was defined as a cluster containing infections from at least two provinces. Patients in interprovincial clusters indicated that they had interprovincial transmission. The interprovincial transmission rate was calculated based on the number of patients in the interprovincial cluster divided by the total sample size. The map was drawn by ArcMap 10.2. Sankey diagram, which is a specific type of flowchart in which the width of the extended branches corresponds to the size of the data flow. In this study, the branch width of the Sankey diagram is used to represent the number of links between Hefei and other provinces in the molecular network. Sankey diagram was drawn using the networkD3 package in R.

### 4.5. Estimating the Effective Reproductive Number (Re)

The Re of each subtype was estimated using the birth–death skyline (BDSKY) serial model in BEAST v2.6.0. Re is calculated as the median ratio of birth and death rates. The BDSKY model employs a piecewise constant birth–death-sampling process to compute the probability density of a phylogeny. In this model, a branching event in the sample tree corresponds to a “birth”, each tip in the tree corresponds to a sampling event, and a death is an unobserved event, i.e., an unsampled recovery or death. Each of these three event types occurs with its own characteristic rate in each interval of the piecewise function. This enables the estimation of epidemiological parameters such as Re. [38,39]. We used the bdskytools package in R to plot the results of BDSKY analyses. Re is defined as the average number of secondary infections caused by an infected person at a specific point in time during an epidemic, when the susceptibility of the population decreased. This value is often used to describe temporal changes of an epidemic in a population, and the Re greater than 1 indicates the growth of an epidemic.

## Figures and Tables

**Figure 1 pathogens-10-01554-f001:**
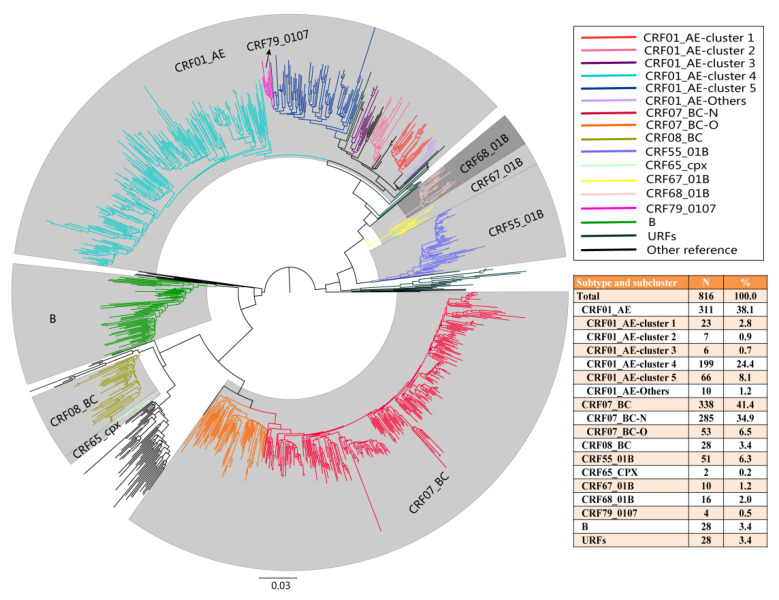
Maximum-likelihood phylogenetic tree of 816 newly reported people with HIV in Hefei from 2017 to 2020. The nucleotide substitution model was GTR + G + I, and support values of the nodes were calculated with a Shimodaira Hasegawa-like test. Clusters with a support value higher than 0.90 (90%) were defined as the same subtype or subclusters. Clades of different colors represent different subtypes and subclusters.

**Figure 2 pathogens-10-01554-f002:**
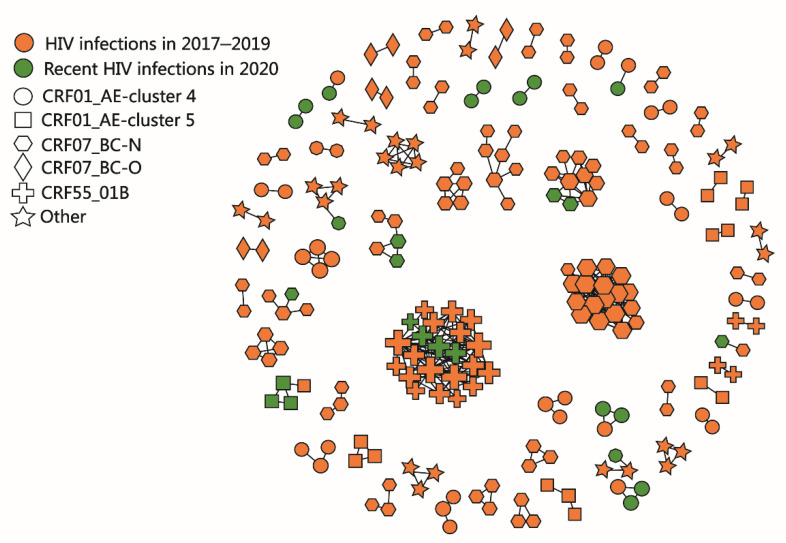
The molecular network diagram of newly reported people with HIV in Hefei from 2017 to 2020. The colors represent a recent infection or past infection and the shapes represent subtypes and subclusters. “Other” group includes other CRFs and URFs.

**Figure 3 pathogens-10-01554-f003:**
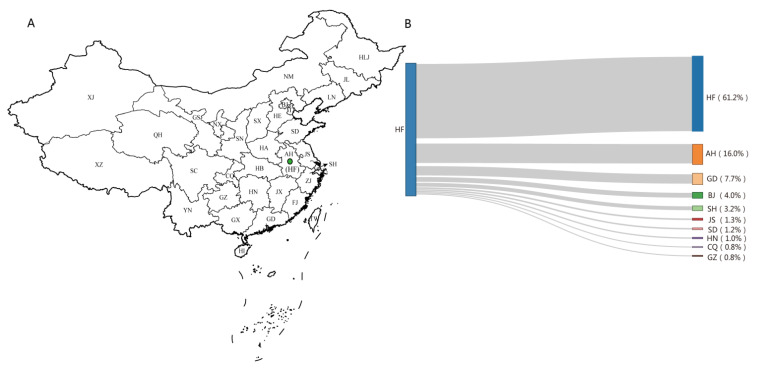
(**A**) The map of China shows the location of the provinces. The green dots in the map represent Hefei (HF). Names of provinces: Anhui (AH), Beijing (BJ), Chongqing (CQ), Fujian (FJ), Guangdong (GD), Gansu (GS), Guangxi Zhuang Autonomous Region (GX), Guizhou (GZ), Henan (HA), Hubei (HB), Hebei (HE), Hefei (HF), Hainan (HI), Heilongjiang (HLJ), Hunan (HN), Jilin (JL), Jiangsu (JS), Jiangxi (JX), Liaoning (LN), Inner Mongolia Autonomous Region (NM), Ningxia Hui Autonomous Region (NX), Qinghai (QH), Sichuan (SC), Shandong (SD), Shanghai (SH), Shaanxi (SN), Shanxi (SX), Tianjin (TJ), Xinjiang Uygur Autonomous Region (XJ), Tibet Autonomous Region (XZ), Yunnan (YN), Zhejiang (ZJ). (**B**) The Sankey diagram of Hefei’s links to other provinces and cities in the molecular network. GX (0.5%), JX (0.5%), SX (0.4%), HA (0.3%), LN (0.2%), TJ (0.2%), SC (0.2%), SN (0.2%), YN (0.2%), ZJ (0.2%), HB (0.1%), HE (0.1%), GS (0.1%) and JL (0.1%) are not listed in the figure.

**Figure 4 pathogens-10-01554-f004:**
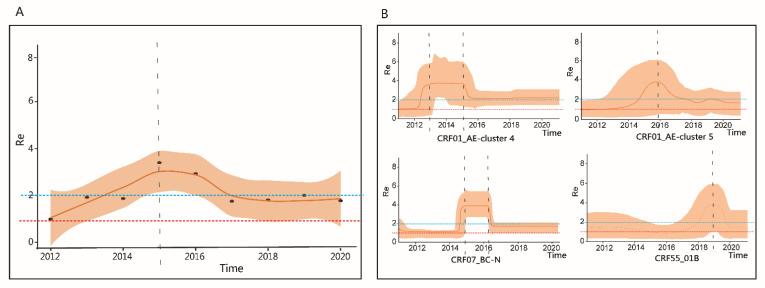
Effective reproductive number (Re) estimates obtained using the birth–death skyline model. (**A**) Weighted Re estimates for each year. (**B**) Re estimates of different subtypes and subclusters. The shaded area represents the 95% Bayesian credible interval. The horizontal red dotted line represents the epidemiological threshold (Re = 1). The horizontal blue dotted line represents the value of Re, which is 2. The horizontal black dotted line represents the time period with the highest Re value.

**Table 1 pathogens-10-01554-t001:** Demographic characteristics of newly reported people with HIV in Hefei from 2017 to 2020.

Characteristic	N	%
Total	816	100.0
Age (years)		
18–29	318	39.0
30–49	247	30.3
≥50	121	14.8
Unknown	130	15.9
Sex		
Male	627	76.8
Female	59	7.2
Unknown	130	15.9
Route of HIV infection		
MSM	505	61.9
HET	178	21.8
IDU	3	0.4
Unknown	130	15.9
Marital status		
Single	379	46.4
Married	210	25.7
Divorced or widowed	97	11.9
Unknown	130	15.9
Ethnicity		
Han	680	83.3
Others	6	0.7
Unknown	130	15.9
Education		
Illiteracy or primary school	81	9.9
Junior middle school	143	17.5
High school and above	462	56.6
Unknown	130	15.9
CD4 count before ART (cells/μL)		
<200	214	26.2
200–350	196	24.0
350–500	140	17.2
>500	136	16.7
Unknown	130	15.9
Year		
2017	66	8.1
2018	162	19.9
2019	215	26.3
2020	373	45.7

**Table 2 pathogens-10-01554-t002:** Comparison of clustering within different subtypes and subclusters of Hefei.

Subtype and Subcluster	N	Cluster (%)	OR (*95% CI*)	*p*	AOR * (*95% CI*)	*p **
Total	816	292 (35.8)				
CRF01_AE-cluster 4	199	54 (27.1)	1.00			
CRF01_AE-cluster 5	66	23 (34.8)	1.44 (0.79–2.60)	0.233	1.38 (0.75–2.52)	0.303
CRF07_BC-N	285	129 (45.3)	2.22 (1.50–3.28)	<0.001	2.41 (1.61–3.61)	<0.001
CRF07_BC-O	53	13 (24.5)	0.87 (0.43–1.76)	0.703	0.95 (0.45–1.98)	0.886
CRF08_BC	28	10 (35.7)	1.49 (0.65–3.43)	0.347	1.51 (0.64–3.61)	0.350
CRF55_01B	51	30 (58.8)	3.84 (2.02–7.27)	<0.001	4.02 (2.10–7.70)	<0.001
B	28	2 (7.1)	0.21 (0.05–0.90)	0.036	0.21 (0.05–0.95)	0.043
Others ^#^	106	31 (29.2)	1.11 (0.66–1.87)	0.696	1.17 (0.68–2.00)	0.572

^#^ Others included CRF01_AE-cluster 1, CRF01_AE-cluster 2, CRF01_AE-cluster 3, CRF01_AE-Others, CRF67_01B, CRF68_01B, CRF79_0107 and URFs. * Adjusted for multivariate logistic regression: age, sex, route of HIV infection, marital status, ethnicity, education, CD4 count before ART and year. OR, odds ratio. AOR, adjusted odds ratio.

**Table 3 pathogens-10-01554-t003:** Comparison of occurring interprovincial transmission within different subtypes and subclusters of Hefei.

Subtype and Subcluster	N	Interprovincial Transmission (%)	OR (*95% CI*)	*p*	AOR * (*95% CI*)	*p **
Total	816	76 (9.3)				
CRF01_AE-cluster 4	199	6 (3.0)	1.00			
CRF01_AE-cluster 5	66	5 (7.6)	2.64 (0.78–8.94)	0.120	2.78 (0.80–9.66)	0.109
CRF07_BC-N	285	51 (17.9)	7.01 (2.95–16.69)	<0.001	6.62 (2.73–16.05)	<0.001
CRF07_BC-O	53	2 (3.8)	1.26 (0.25–6.44)	0.780	1.70 (0.32–9.14)	0.538
CRF08_BC	28	1 (3.6)	1.19 (0.14–10.28)	0.874	1.55 (0.17–13.97)	0.699
CRF55_01B	51	6 (11.8)	4.29 (1.32–13.92)	0.015	5.71 (1.70–19.15)	0.005
B	28	1 (3.6)	1.19 (0.14–10.28)	0.874	1.29 (0.14–11.51)	0.821
Others ^#^	106	4 (3.8)	1.26 (0.35–4.57)	0.724	1.54 (0.42–5.70)	0.520

^#^ Others included CRF01_AE-cluster 1, CRF01_AE-cluster 2, CRF01_AE-cluster 3, CRF01_AE-Others, CRF67_01B, CRF68_01B, CRF79_0107 and URFs. * Adjusted for multivariate logistic regression: age, sex, route of HIV infection, marital status, ethnicity, education, CD4 count before ART and year. OR, odds ratio. AOR, adjusted odds ratio.

## Data Availability

Data are contained within the article.

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
