# Peer review of "Epidemiology and Molecular Transmission Characteristics of HIV in the Capital City of Anhui Province in China"

_pathogens, 2021, doi:10.3390/pathogens10121554_

Round 1

Reviewer 1 Report

Zheng et al, in their paper entitled “Epidemiology and molecular transmission characteristics of HIV in the capital city of Anhui province in China”, analyzed 816 newly reported HIV infections from 2017-2020 in Hefei, Anhui Province, China.  Sequencing analysis identified 4 major subtypes (CRF07_BC-N, CRF01_AE-cluster 4, CRF01_AE-cluster 5 and CRF55_01B ) and further molecular network analysis indicated the rapid spread of two strains, CRF55_01B and CRF07_BC-N, with high interprovincial transmission rates.  In addition, the Re in Hefei was notably high, especially with CRF55_01B estimated at 2-4 in 2018 and 2019.   These findings help in identifying HIV strains and the spread of HIV in Hefei, which is important for prevention control strategies.

Paper is well written, and I would offer the following suggestions:

  1. Table 1. Since everything in the “Characteristic” column is center justified, it makes it hard to discern each sub-category (i.e. Age, Sex, Route of Infection, Marital status, etc). Either bold or left justify the sub-category title to make it easier for the reader.

  1. Line 95: Define for the reader the LAG-Aviditiy EIA test. What is means; what the test tells you.

  1. Figure 2. It is hard to discern the “circle” from the “jagged circle” for Other in the diagram. Not sure if this can be fixed.

  1. Table 2/Table 3. Define the abbreviations “OR” and “AOR”.

  1. Figure 3. Define or spell out for those of us that do not live in China, the province/city names. JX? HE? SN? etc….   It might be interesting to insert a map here showing where each province/city lies in relation to Hefei since you are talking about provincial transmission rates.

  1. Pol sequences were generated for the 816 patients. Please submit to Genbank and provide accession numbers to share the sequence information gathered.  Or provide a Supplemental FASTA file with the sequences.

  1. Line 142. Figure 4 title is mislabeled as Figure 1.

     8.  Of interest.  It would be interesting if the 2020 active HIV transmission clusters show in Figure 2 were connected to COVID-19 city-lockdowns.  Limiting travel, HIV transmission clusters would likely be limited to the location of the infected individual and be more pronounced. 

Author Response

Response to Reviewer 1 Comments

First of all, thank you very much for your valuable suggestions on my article, which made my article more perfect as a whole. I will give my response to your questions one by one.

Point 1: Table 1. Since everything in the “Characteristic” column is center justified, it makes it hard to discern each sub-category (i.e. Age, Sex, Route of Infection, Marital status, etc). Either bold or left justify the sub-category title to make it easier for the reader.

Response 1: We have left justified the sub-category title of Table 1.

Point 2: Line 95: Define for the reader the LAG-Avidity EIA test. What is means; what the test tells you.

Response 2: Line 267-270: We have added the description of what is the LAG-Avidity EIA test in the section of materials and methods. “It includes preliminary screening test and confirmation test. Samples with ODn less than or equal to 2.0 in the preliminary screening test shall be validated. If the ODn value of the test sample is confirmed to be between 0.4 and 1.5, the sample is recently infected.”

Point 3: Figure 2. It is hard to discern the “circle” from the “jagged circle” for other in the diagram. Not sure if this can be fixed.

Response 3: We have replaced the “jagged circle” with the “Pentagram” in Figure 2.

Point 4: Table 2/Table 3. Define the abbreviations “OR” and “AOR”.

Response 4: We have defined the abbreviations “OR” and “AOR” in Table 2 and Table 3. “OR, odds ratio. AOR, adjusted odds ratio.”

Point 5: Figure 3. Define or spell out for those of us that do not live in China, the province/city names. JX? HE? SN? etc….   It might be interesting to insert a map here showing where each province/city lies in relation to Hefei since you are talking about provincial transmission rates.

Response 5: Line 137-145: We have inserted a map which shows the location of each province and spelled out the province/city names in Figure 3. “Anhui (AH), Beijing (BJ), Chongqing (CQ), Fujian (FJ), Guangdong (GD), Gansu (GS), Guangxi Zhuang Autonomous Region (GX), Guizhou (GZ), Henan (HA), Hubei (HB), Hebei (HE), Hefei (HF), Hainan (HI), Heilongjiang (HLJ), Hunan (HN), Jilin (JL), Jiangsu (JS), Jiangxi (JX), Liaoning (LN), Inner Mongolia Autonomous Region (NM), Ningxia Hui Autonomous Region (NX), Qinghai (QH), Sichuan (SC), Shandong (SD), Shanghai (SH), Shaanxi (SN), Shanxi (SX), Tianjin (TJ), Xinjiang Uygur Autonomous Region (XJ), Tibet Autonomous Region (XZ), Yunnan (YN), Zhejiang (ZJ).”

Point 6: Pol sequences were generated for the 816 patients. Please submit to Genbank and provide accession numbers to share the sequence information gathered.  Or provide a Supplemental FASTA file with the sequences.

Response 6: We are very sorry that these sequences of this study are available from the corresponding author upon reasonable request.

Point 7: Line 142. Figure 4 title is mislabeled as Figure 1.

Response 7: We have changed the label “Figure 4”.

Point 8: Of interest.  It would be interesting if the 2020 active HIV transmission clusters show in Figure 2 were connected to COVID-19 city-lockdowns. Limiting travel, HIV transmission clusters would likely be limited to the location of the infected individual and be more pronounced.

Response 8: Your proposal is very good. We have added a little discussion about this in Line 243-245. However, we cannot confirm whether the active HIV transmission clusters in this study is related to COVID-19 city-lockdowns, because we do not have detailed movement trajectory of these people. We can look into that in future studies.

Finally, thank you again for all your suggestions. These are of guiding significance to my thesis writing and scientific research work.

Reviewer 2 Report

Review Pathogens-1461114

Line 15 Jiangsu instead of jiangsu

Lines 19-20 and line 155 CRF07_BC-N (34.9%), CRF01_AE-cluster 4 (24.4%), CRF01_AE-19 cluster 5 (8.1%). There is no sense to describe prevalent subtypes directly linked with clusters, these are two different concepts. I will write “The most prevalent subtypes were CRF07_BC (34.9%), CRF01_AE (32.5%) and CRF55_01B (6.3%).

Line 65 : the sampling ratio was 55.6% (816/1468). What is the sampling ratio?

Line 84 : The HIV pol fragments of 28 cases of URF were found to be the recombinant 84 strains of CRF01_AE and B or C subtype by JPHMM and RIP. It’s not clear that these URF were in the same branch in the tree figure 1.

Line 95 : A total of 816 plasma samples were eligible for LAg-Avidity EIA, and 22.1% (180/816) 95 of them were positive. What is your conclusion????

Line 119 : The links between Hefei and each province are shown in Figure 3. Figure 3. The Sankey diagram. This part is not clear at all. In the Materials and Methods, you should described your method and the diagram more precisely.

Line 132 : Estimating the Re. How do you manage to determine “the average number of secondary infec-268 tions caused by an infected person at a specific point in time during an epidemic” in your study? Clarify this point.

Line 154 : Therefore, especially in large cities, it is necessary to strengthen the intervention 154 among MSM. What intervention do you mean? Prevention and diagnosis?

Lines 180-181 : same remark as lines 19-20

Line 215 : this study used people with HIV. It will be better to write, “this study involved people with HIV”

Lines 229-230 : “patients with HIV infection who had not received any antiviral treatment from January 2017 to December 2020”. Why is there any resistance analyze because this population is suitable for resistance transmission study also.

Line 243 : WHO 243 sequence quality control was performed to obtain qualified sequences. Which WHO quality control, give reference.

Line 249 : 4.4. Phylogenetic analysis and HIV molecular network construction. There is no definition of cluster characteristics used in figure 1. For example, why don’t you describe the B cluster? Have they got all the same recombination breakpoints? Is it a new CRF?

Author Response

Response to Reviewer 2 Comments

First of all, thank you very much for your valuable suggestions on my article, which made my article more perfect as a whole. I will give my response to your questions one by one.

Point 1: Line 15 Jiangsu instead of jiangsu.

Response 1: We have replaced “jiangsu” with “Jiangsu”.

Point 2: Lines 19-20 and line 155 CRF07_BC-N (34.9%), CRF01_AE-cluster 4 (24.4%), CRF01_AE-19 cluster 5 (8.1%). There is no sense to describe prevalent subtypes directly linked with clusters, these are two different concepts. I will write “The most prevalent subtypes were CRF07_BC (34.9%), CRF01_AE (32.5%) and CRF55_01B (6.3%).

Response 2: We have revised the description. “The most prevalent subtypes were CRF07_BC (41.4%), CRF01_AE (38.1%) and CRF55_01B (6.3%).”

Point 3: Line 65: the sampling ratio was 55.6% (816/1468). What is the sampling ratio?

Response 3: Line 67-68: We have added the description of this section. “According to the local center for Disease Control and Prevention, there were 1,468 newly reported HIV infections in Hefei from 2017 to 2020.” The sampling ratio is equal to the sample size included in this study divided by the total sample size.

Point 4: Line 84: The HIV pol fragments of 28 cases of URF were found to be the recombinant strains of CRF01_AE and B or C subtype by JPHMM and RIP. It’s not clear that these URF were in the same branch in the tree figure 1.

Response 4: In figure 1, the dark green branches are URFs, these URFs are in different branches.

Point 5: Line 95: A total of 816 plasma samples were eligible for LAg-Avidity EIA, and 22.1% (180/816) 95 of them were positive. What is your conclusion????

Response 5: Line 102-104 and Line 268-271: We have revised our conclusions. “and 22.1% (180/816) of them were recently infected.” And we added the definition and purpose of the LAg-Avidity EIA in the section of materials and methods (4.2. Laboratory tests). “Rent infections were detected using HIV LAg-Avidity EIA [14]. It includes preliminary screening test and confirmation test. Samples with ODn less than or equal to 2.0 in the preliminary screening test shall be validated. If the ODn value of the test sample is confirmed to be between 0.4 and 1.5, the sample is recently infected.”

Point 6: Line 119: The links between Hefei and each province are shown in Figure 3. Figure 3. The Sankey diagram. This part is not clear at all. In the Materials and Methods, you should describe your method and the diagram more precisely.

Response 6: Line 301-305: We have added description in the section of materials and methods (4.4. Phylogenetic analysis and HIV molecular network construction). Sankey diagram is a specific type of flowchart in which the width of the extended branches corresponds to the size of the data flow. In this study, the branch width of Sankey diagram is used to represent the number of links between Hefei and other provinces in the molecular network. Sankey diagram was drawn using the networkD3 package in R.”

Point 7: Line 132: Estimating the Re. How do you manage to determine “the average number of secondary infections caused by an infected person at a specific point in time during an epidemic” in your study? Clarify this point.

Response 7: Line 308-315: We have added description in the section of materials and methods (4.5. Estimating the effective reproductive number (Re). “Re is calculated as the median ratio of birth and death rates. The BDSKY model employs a piecewise constant birth-death-sampling process to compute the probability density of a phylogeny. In this model, a branching event in the sample tree corresponds to a “birth’’, each tip in the tree corresponds to a sampling event, and a death is an unobserved event, i.e. an unsampled recovery or death. Each of these three event types occurs with its own characteristic rate in each interval of the piecewise function. This enables the estimation of epidemiological parameters such as Re.” The specific calculation method is described in detail in reference 19.

Point 8: Line 154: Therefore, especially in large cities, it is necessary to strengthen the intervention among MSM. What intervention do you mean? Prevention and diagnosis?

Response 8: Line 173-176: We have added description that how we can intervene MSM. “The government and relevant departments can raise the awareness of MSM to reject high-risk behaviors and encourage them to perform pre-exposure prophylaxis (PrEP) and post-exposure prophylaxis (PEP) through publicity and education.”

Point 9: Lines 180-181: same remark as lines 19-20

Response 9: Line 176-177: We have revised the description. “the main prevalent subtypes were CRF07_BC, CRF01_AE and CRF55_01B.”

Point 10: Line 215: this study used people with HIV. It will be better to write, “this study involved people with HIV”

Response 10: We have replaced “used” with “involved”.

Point 11: Lines 229-230: “patients with HIV infection who had not received any antiviral treatment from January 2017 to December 2020”. Why is there any resistance analyze because this population is suitable for resistance transmission study also.

Response 11: Your proposal is very good. In fact, we have done research on drug resistance at the same time, but this article is not focused on drug resistance, so it is not included in this article. It will be presented in another article.

Point 12: Line 243: WHO sequence quality control was performed to obtain qualified sequences. Which WHO quality control, give reference.

Response 12: Line 275-276: We have added the web address of “WHO HIVDR QC TOOL” (https://sequenceqc.bccfe.ca/who_qc).

Point 13: Line 249: 4.4. Phylogenetic analysis and HIV molecular network construction. There is no definition of cluster characteristics used in figure 1. For example, why don’t you describe the B cluster? Have they got all the same recombination breakpoints? Is it a new CRF?

Response 13: I don't know if I understand correctly, I guess you mean figure 2 instead of figure 1.

Line 113-114: We have added the grouping explanation under figure 2. Other CRFs and URFs account for less and they are not the focus of our study, so they are grouped together in the “Other” group. Subtype B is not in the network under the threshold of 0.5% genetic distance.

Finally, thank you again for all your suggestions. These are of guiding significance to my thesis writing and scientific research work.

Round 2

Reviewer 2 Report

Dear Authors,
You clearly improved your manuscript and in my point of view, it is really better now. I wish you the best for your future.